# Decreased tourism during the COVID-19 pandemic positively affects reef fish in a high use marine protected area

Kevin C. Weng[1]*, Alan M. Friedlander[2,3], Laura Gajdzik[4], Whitney Goodell[2,3], Russell T. Sparks[4]

**1** Virginia Institute of Marine Science, William & Mary, Gloucester Point, Virginia, United States of America, **2** Department of Biology, Fisheries Ecology Research Lab, University of Hawai'i at Mānoa, Honolulu, Hawai'i, United States of America, **3** Pristine Seas, National Geographic Society, Washington, District of Columbia, United States of America, **4** Division of Aquatic Resources, Department of Land and Natural Resources, Honolulu, Hawai'i, United States of America

* kcweng@wm.edu

**Data Availability Statement:** Human abundance and fish biomass data will be deposited in

## Abstract

Humans alter ecosystems through both consumptive and non-consumptive effects. Consumptive effects occur through hunting, fishing and collecting, while non-consumptive effects occur due to the responses of wildlife to human presence. While marine conservation efforts have focused on reducing consumptive effects, managing human presence is also necessary to maintain and restore healthy ecosystems. Area closures and the tourism freeze related to the COVID-19 pandemic provided a unique natural experiment to measure the effects of decreased tourism on fish behavior in a high use no-take marine protected area (MPA) in Hawai'i. We found that when tourism shut down due to COVID restrictions in 2020, fish biomass increased and predatory species increased usage of shallow habitats, where tourists typically concentrate. When tourism resumed, fish biomass and habitat use returned to pre-pandemic levels. These displacement effects change fish community composition and biomass, which could affect key processes such as spawning, foraging and resting, and have knock-on effects that compromise ecosystem function and resilience. Managing non-consumptive uses, especially in heavily-visited MPAs, should be considered for sustainability of these ecosystems.

## Introduction

Non-consumptive effects of humans are well documented in both terrestrial [1], and marine systems [2]. Non-consumptive effects are generated through either avoidance of or attraction to human activities and can induce behavioral change, displacement, habituation, crowding, and dietary impacts [3], and disruptions in foraging, reproduction [4], and resting [5, 6]. Tourism impacts in marine systems have been researched extensively with respect to shark and fish provisioning [7]. Effects of human presence have been assessed for marine mammals, sharks, birds, turtles [2] and in freshwater systems [8]. Boat noise is known to impact fish stress and communication [9], predation mortality [10] and larval settlement [11].

datadryad. Fish occupancy data will be deposited in the PIRAT node of the Ocean Tracking Network. https://piratnetwork.org Fish tagging was conducted under an animal care protocol (IACUC-2020-10-05-14561-kcweng) approved by the William & Mary Animal Care and Use Committee (IACUC), USDA 52-R-0002, OLAW D16-00419 #A3713-01, and a Special Activity Permit from the State of Hawaiʻi (2021-28)."

**Funding:** This study was funded through contributions from private citizens facilitated by the Maui Nui Marine Resources Council; a grant from the State of Hawaiʻi Department of Land and Natural Resources to VIMS (PO C10740); and the U.S. Fish and Wildlife Service Sport Fish Restoration (Dingell-Johnson) Program through the Hawaiʻi DLNR Division of Aquatic Resources' Marine Fisheries Survey program (F21AF01491). The funders had no role in study design, data collection and analysis, decision to publish, or preparation of the manuscript.

**Competing interests:** The authors have declared that no competing interests exist.

Less research has been conducted to understand human presence effects on marine fishes [8]. In marine systems, fishes often perceive humans as predators and avoid them, so the resulting alterations in fish distribution are non-consumptive effects [12], in contrast to the consumptive effect of fishing. In prior studies, fish diversity and biomass declined temporarily in the presence of snorkelers [13], and the long-term disturbance effects of ongoing human presence [12, 14–16] caused habitat shifts at the cost of reduced access to resources [17]. Fish community structure was altered at intensive tourism sites [14, 15] and increased when human activities were reduced [18].

Non-consumptive effects can be direct, when human presence displaces a fish from a habitat, or indirect, when human presence displaces a predatory fish, thus releasing prey species from risk [12, 19]. Since herbivores avoid high-risk foraging locations when predators are abundant, indirect effects can alter community composition and abundance of primary producers [20]. Snorkeling and diving are concentrated in shallower waters [19] so these non-consumptive effects are strongest near shore [21, 22].

Tourism is the main driver of non-consumptive human effects in protected areas and is one of the largest and fastest-growing sectors of the global economy. This growth comes with both positive and negative ecological impacts [23]. Tourism in protected areas generates large economic benefits, contributing to the management of the protected areas as well as local and travel economies [24, 25]. While tourism can facilitate the preservation of natural resources by increasing the economic value of living animals and intact habitats [26], it can lead to degradation of the resources that tourists are paying to see [27]. Different levels of protection in marine protected areas (MPAs) lead to divergent conservation outcomes, and even non-extractive uses such as tourism can have profound impacts [28].

Since tourism can have negative impacts [28], large reductions in human activity are expected to allow wildlife to reestablish optimal habitat use [17], resulting in higher biomass particularly at higher trophic levels. The COVID-19 pandemic caused a substantial decline in tourism [29]. Global economic losses in the travel and tourism sector are in the range of USD 4 to 12 trillion in gross domestic product, 164 to 514 million jobs, and USD 363 to 1134 billion in capital investment [30]. In French Polynesia, the absence of tourism during a 45 day lockdown resulted in a short term increase in fish biomass [31]. In Hawaiʻi the pandemic caused a 71% decrease in visitors, with 2.7 million in 2020, as compared to 10.4 million in 2019 [32], so a resurgence of wildlife at tourist sites was possible.

In response to ongoing environmental degradation, the State of Hawaiʻi is developing a marine resource management initiative (https://dlnr.hawaii.gov/holomua/), in the context of the United Nations Convention on Biological Diversity. Hawaiʻi receives approximately 10 million tourists annually, many of whom participate in marine tourism [33], so management of tourism impacts is a priority [22]. Managers must set ecological performance targets and then achieve them by controlling impacts. At a given level of economic benefit, ecological impacts are likely to be lower when tourism focuses on premium experiences with fewer visitors, each spending more money, instead of minimizing costs to maximize visitor numbers [34]. By shifting towards higher margin tourism, economic benefits are maintained while decreasing the number of visitors [2, 35].

In Hawaiʻi, the State designated Marine Life Conservation Districts (MLCDs) as no-take MPAs to conserve and replenish marine resources [36], but these MPAs are frequently visited by a high number of tourists, and non-consumptive effects remain to be quantified. One of these highly visited MPAs is the Molokini Shoal MLCD. Despite its small size (0.36 km$^2$), Molokini received an average of >1,000 visitors per day over the past two decades [37]. In the 1970s, tour operators became concerned about fishing impacts at Molokini Crater and advocated for the creation of an MPA, which was officially designated in 1977 (Hawaiʻi

Administrative Rules 13–31). The Molokini Shoal MLCD created a two-zone management system that prohibited all fishing inside the crater, and allowed only trolling in the zone outside of the crater. As the number of tours increased, operators voiced concern about the impacts of tourism volume at the MLCD, as well as damage to corals caused by tour boat anchors. In 1987, a limited-entry permit system was established, allowing 42 tour vessels to visit the MLCD, and in 1995 anchoring was prohibited and moorings were installed for both commercial and private users [37]. In 2014 a conservation action plan was created, highlighting four priorities: the coral reef ecosystem, apex predators, seabirds, and place-based nature experiences for visitors [37].

To make informed decisions, managers require data on the effects of human activity on wildlife and the ecosystem as a whole. For Molokini, the tour operator permit system includes mandatory reporting of vessels, passenger numbers, and activities to the Maui Division of Aquatic Resources (DAR). The State of Hawaiʻi conducts regular monitoring of fishes and benthic resources within the MLCD, while other scientists have conducted in-depth studies of the area [38, 39]. Friedlander et al. quantified the 'reserve effect' of Hawaiʻi's MPAs by examining fish biomass inside these MPAs compared to adjacent areas and found that Molokini had among the highest reserve effects in the state, with biomass more than six times greater inside the MLCD compared to adjacent areas [22]. Filous et al. [40] studied predator movement patterns and found that *Caranx melampygus*, a prized sportfish species, was displaced from inside the MLCD crater during periods of high human and vessel abundance.

In this study, we build on prior research by using the COVID pandemic as a perturbation in human abundance in a high-use MPA. We hypothesize that fish biomass and habitat use are negatively correlated with human abundance. The number of visitors to Molokini before and after the COVID-19 pandemic was quantified using data from mandatory vessel logbooks, and was examined in relation to the biomass of reef fishes within the MLCD and the movement patterns of predatory fishes. Insights gained from this study will inform management planning and are relevant to three of the four priorities in the Molokini conservation action plan (preserve the coral reef ecosystem and predators, and provide place-based nature experiences). The study sheds light on the non-consumptive effects of tourism on marine wildlife.

## Materials and methods

### Study area

Molokini is a small, crescent-shaped islet located in the ʻAlalākeiki Channel, 4.2 km off the south coast of Maui (Hawaiʻi, U.S.). It is the remnant of the rim of a basalt tuff crater, located between Maui and Kahoʻolawe [41]. Due to its distance from the coast of Maui it does not receive runoff and sedimentation from land. The interior of the crater is managed as a no-take zone (0.16 km$^2$), while the water surrounding the outside of the crater (0.19 km$^2$) is managed as a trolling-only zone, where other fishing methods are prohibited. The majority of tourists who visit the MLCD use the interior zone, with snorkeling being the primary activity, while use of the outside zone is much lower, primarily by scuba divers (DAR logbook data).

### Visitor abundance

Detailed reporting by permitted tour operators was initiated in 2012 and is maintained in a vessel logbook dataset by DAR. This dataset does not cover private vessels, which are usually small boats carrying four or fewer persons, and account for a small minority of people visiting the MLCD. Reporting in the vessel log book includes the vessel name, permit number, date, arrival and departure times, location and number of mooring buoy(s) visited, and the numbers of participants in each underwater activity(scuba, snuba, snorkel, and other). In downstream analyses,

we used the maximum number of participants from all activities in each vessel for each day and each month as our proxy of monthly human abundance for the period of 2013–2021.

### *In situ* fish surveys

Diver surveys measured fishes at all trophic levels within the MLCD using two methods. The Fish Habitat Utilization Surveys (FHUS) method was initially conducted in 2004 by the Fisheries Ecology Laboratory from the Hawai'i Institute of Marine Biology as part of an evaluation of MPAs across the State. This method was repeated in 2020 and 2021 to assess the impacts of COVID on fishes in the MLCD. he Fish and Habitat Utilization Survey (FAHU) method started in 2018 and is part of the marine monitoring program of State of Hawai'i (DAR). The FAHU method is conducted every year in August/September but additional surveys were done in April 2020, September 2020, and April 2021 for this study.

The FHUS and FAHU methods follow a similar procedure. A scuba diver swam a 25 x 5 m transect at a constant speed, identified all fishes visible within 2.5 m to either side of the centerline (125 $m^2$ transect area) to the lowest possible taxon, and estimated the total length (TL) of fishes to the nearest centimeter. Swimming duration varied from 10–15 min, depending on habitat complexity and fish abundance. FAHU and FHUS differ in the site selection and number of transects. The FHUS method consists of 23 transects that are selected using a spatially explicit stratified random sampling design [38], whereas the FAHU method includes 40 randomly selected transects over contiguous, complex shallow (≤8 m deep) and deep (≥9–16 m deep) aggregate reefs with relatively high coral cover. To ensure consistency within surveys, FHUS dives were conducted by AMF and WG, while FAHU surveys were conducted by the DAR survey team directed by RTS.

Length estimates of fishes from visual censuses were converted to weight using the following length-weight relationship: $W = aSL^b$ where the parameters a and b are constants for the allometric growth equation and SL is standard length in mm and W is weight in grams. Total length was converted to standard length (SL) by multiplying standard length to total length-fitting parameters obtained from FishBase [42].

### Fish habitat use and occupancy

We used acoustic telemetry to follow the movements of fishes within the Molokini MLCD, keeping consistent with the methodology of a previous study [40]. Given finite resources, we tagged only predatory species since these are involved in both direct and indirect effects. We placed seven passive acoustic monitoring devices (VR2W receiver, Vemco Ltd., Halifax, Nova Scotia) around the outside and inside of Molokini Crater (Fig 1). The monitoring array allowed us to measure presence-absence in the MPA and determine if a fish was inside or outside of the crater. Acoustic monitoring receivers were attached to the benthos using a line, held vertically by a float. Lines were tied around boulders in hard substrate areas and attached to ~ 27 kg cement blocks in sand substrate areas. Receivers were downloaded at approximately 6-month intervals covering May 2020 through May 2021.

Predatory fishes were captured with standard rod and reel fishing gear and tagged with Vemco v13 and v16 coded acoustic tags. Methods are described in detail in Filous et al. [40]. Tags were surgically implanted into the peritoneum of the fish and the incision closed with surgical sutures. Surgical tools and tags were stored in Chlorhexidine prior to the procedure. Fishes were held supine in a padded cradle with saltwater flowing over the gills. Immediately following the surgery each animal was gently returned to the water and released. Fish tagging was conducted under an animal care protocol (IACUC-2020-10-05-14561-kcweng) approved by the William & Mary Animal Care and Use Committee (IACUC, USDA 52-R-0002, OLAW D16-00419 #A3713-01), and a Special Activity Permit from the State of Hawai'i (2021–28).

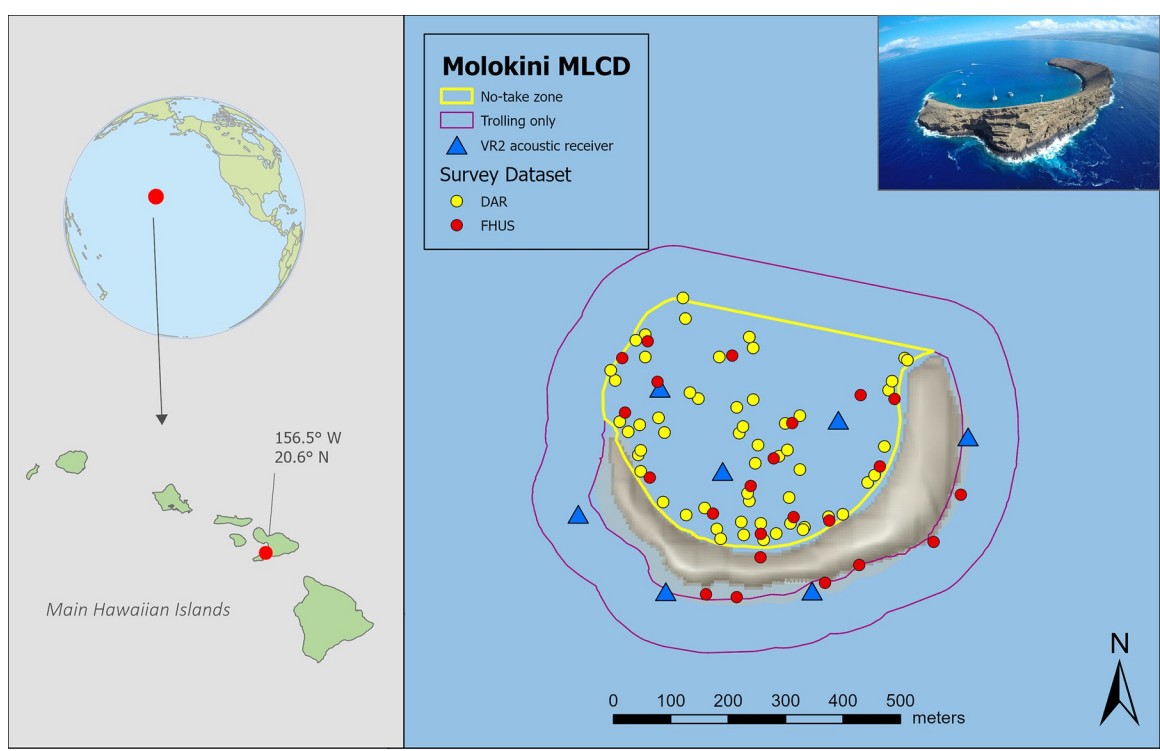

**Fig 1. Map of the study site, Molokini Shoal Marine Life Conservation District, located in Maui, Hawaii, USA.** Blue triangles show locations of acoustic tracking receivers, yellow and red dots show locations of fish surveys, yellow line shows the boundary of the no-fishing zone inside the crater, and red line shows the boundary of the trolling-only zone around the outside of the crater. Hillshade basemap derived from USGS Digital Elevation Model (DEM) [43]. The partially submerged volcanic crater that forms a wall around the interior is shown in the aerial photo ('Molokini Crater' by Bossfrog from Wikimedia Commons under the Creative Commons Attribution-Share Alike 4.0 International license).

## Analysis

**Visitor abundance.**  To explore the variation in human abundance in Molokini across time and detect potential effects of the COVID-19 pandemic, we used generalized additive (GAM) models and generalized additive mixed models (GAMM) with the R-package mgcv [44]. These types of model were selected to allow the detection of non-linear patterns [45], which are often present in time-series datasets [46]. We aggregated our human abundance data by month (covering nine years from 2013 to 2021) and the COVID-19 pandemic was defined as lasting from March 2020 (start of lockdowns) to May 2021 (relaxing of passenger restrictions for commercial boats).

We started with the most complex model (i.e., adding COVID as a factor and varying the slope of the models pre- and post-pandemic) and reduced complexity in a stepwise fashion. Smoother functions for the continuous time covariates (i.e., months and times) were fitted with cyclic cubic splines and cubic regression splines, respectively. The number of knots (k) were determined by comparing the estimated degree of freedom to k. The optimal number of knots for months was seven and 12 for times. Number of visitors had a Gaussian distributed error term after trialing different distribution families (e.g., Poisson, quasi-Poisson).

Additionally, we added an autoregressive-moving-average (ARMA) residual autocorrelation structure to account for any dependence in our data. However, neither the autocorrelation

function (ACF) and partial autocorrelation function (pACF) plots nor the auto.arima function from the R-package forecast confirmed the need to add an ARMA. The model with the best fit was selected using Akaike Information Criterion (AIC). When models did not differ by more than 2 AIC, the simplest model was selected. Model assumptions were checked by plotting model residuals against fitted values with the R-package gratia. All analyses were conducted in R [47].

*In situ* **fish surveys.** Fish biomass (i.e., g m$^{-2}$) from both the FHUS and FAHU surveys were compared among sampling periods using a general linear model with a normal distribution and log-link function based on AIC$_c$. Contrasts using -loglikelihoods estimates were used to compare individual sampling periods from one another (2004, 2021, 2022).

Comparisons of fish assemblage structure among sampling periods based on biomass were investigated for both FHUS and FAHU surveys using permutation-based multivariate analysis of variance (PERMANOVA). Bray–Curtis similarity matrices were created from biomass of fish taxa. Prior to analyses, fish biomass was square root transformed. Interpretation of PERMANOVA results was aided using individual analysis of similarities (ANOSIM).

Principal Coordinate Analysis (PCO) examines fish assemblage structures among years. Eigenvectors were superimposed on the PCO plots to displace the relative contribution and direction of influence of taxa to the observed variation among years (Pearson product-moment correlations $\geq$ 0.5). Analysis was conducted in R 4.1.1 [47].

Principal Coordinate Analysis (PCO) was used to display fish assemblage structure among sampling periods. The primary taxa vectors driving the ordination (Pearson correlation Product-uct movement correlations > 0.5) were overlaid on the PCO plots to visualize the major taxa that explained the spatial distribution patterns observed.

**Fish habitat use and occupancy.** Fish tracking data were downloaded from the passive acoustic monitoring array, processed in the manufacturer supplied software (Vue, Vemco Ltd.) and exported to csv. The fish detection data were used as the response variable. Vessel and human abundance data from the DAR database were used as the explanatory variables. Fish habitat utilization was represented by the daily count of detections on the monitoring array for all species, and was also subsetted to detections occurring for *C. melampygus* during morning hours (07:00–11:00) on the inside of the crater (yellow zone on Fig 1), where vessels moor and the majority of people are concentrated. Human abundance was represented by the daily counts of people, vessels, and people per boat. We used a GLM, and since our response and predictor variables were count data with overdispersion, we used the negative bionomial distribution with log link, with the R-package MASS. We assessed collinearity of predictor variables by calculating the variance inflation factors (VIF) with the R-package car. Predictors with VIF>10 were not used together in a model. When models did not differ by more than 2 AIC, the simplest model was selected (see supplement for model formulations). We determined if our selected model was different from the null model using a Chi-square test. Analysis was conducted in R4.1.1 [47].

## Results

### Visitor abundance

Human abundance at Molokini varied substantially over the course of the study with values ranging from zero to 43,511 people monthly (Fig 2). The number of people consistently peaked during the summer every year ($R^2$ = 0.82, EDF = 4.06, p < 0.001) until 2020 when the COVID-19 pandemic occurred (EDF = 4.78 x 10$^{-07}$, p = 0.7). From March to April 2020 the number of visitors declined to zero and remained low through June, July, and August 2020 before rising rapidly (EDF = 2.50, p < 0.001). By June 2021 the human abundance at Molokini reached 37,325 of visitors monthly (Fig 2), returning to pre-pandemic levels.

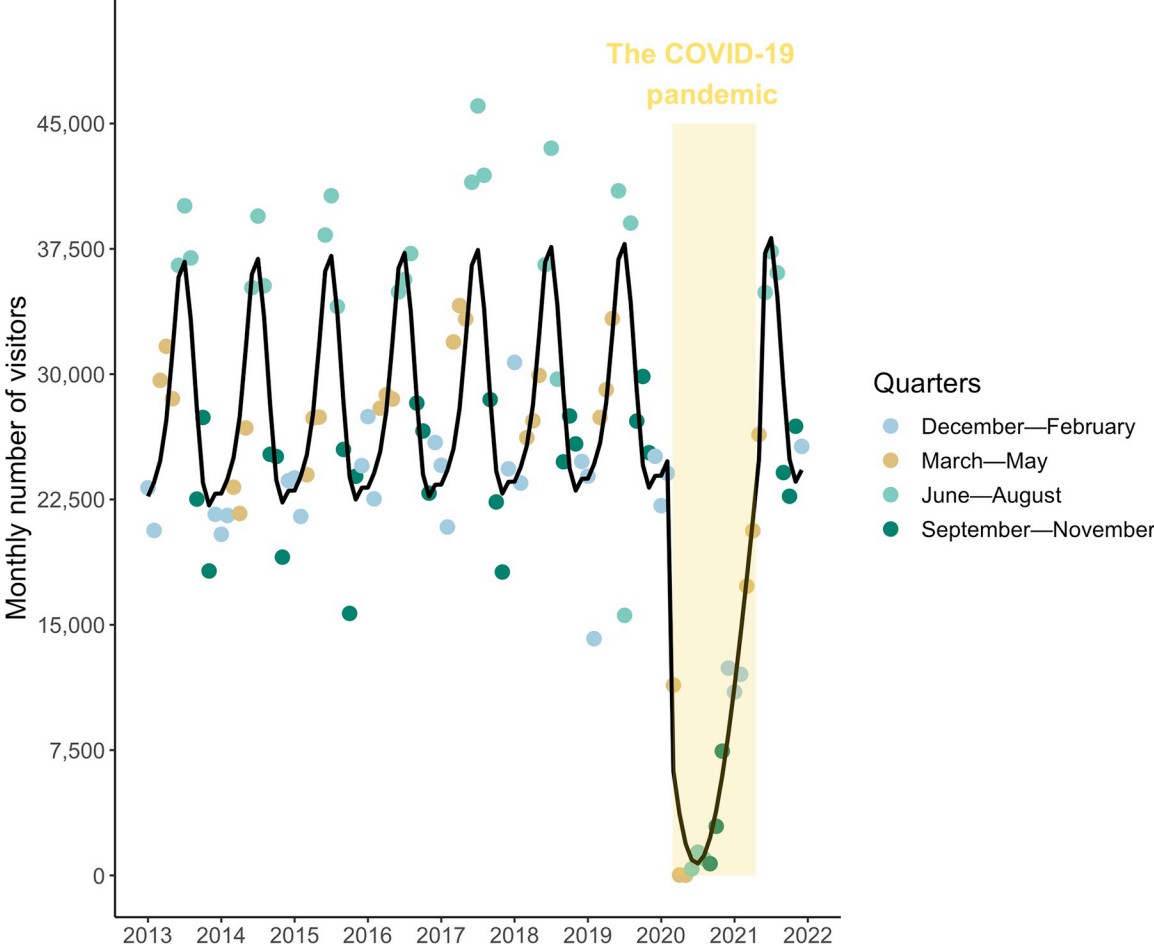

**Fig 2. Monthly variation in the number of people visiting Molokini since 2013.** Circles show monthly counts of people at Molokini and are colored by quarters (q) that each corresponds to a 3-month period. The black line is the result of the GAM model. The yellow box denotes the period of travel restrictions associated with the COVID-19 pandemic (from March 2020 to May 2021).

### *In situ* fish surveys

Based on the FHUS surveys, fish biomass in the Molokini MLCD has declined by 44% over two decades (Fig 3A), which was driven by decreases in predators including giant trevally (*Caranx ignobilis*) and white-tip reef shark *Triaenodon obesus* (Fig 3B). Fish biomass from the long-term surveys (2004, 2022, and 2021) was significantly different among years ($\chi^2 = 4.77$, p = 0.029), with 2021 significantly lower than 2004. Fish assemblage structure based on biomass was significantly different among years (PERMANOVA pseudo-$F_{2,68} = 2.395$, p = 0.001), with 2004 significantly different from 2022 (ANOSIM R = 0.142, p = 0.005) and 2021 (ANOSIM R = 0.159, p = 0.001). The years 2020 and 2021 were not significantly different from one another (ANOSIM R = 0.012, p = 0.304).

Fish biomass from the DAR surveys from January 2018 to April 2021 was significantly different among sampling periods ($\chi^2 = 19.18$, p = 0.002), with biomass during the April 2020 sampling period significantly different from all other sampling periods (Fig 3C). All other sampling periods were not significantly different from one another. Fish assemblage structure based on biomass was significantly different among years (PERMANOVA pseudo-$F_{5,236} = 2.37$, p = 0.001). Most sampling periods were significantly different from one another (all

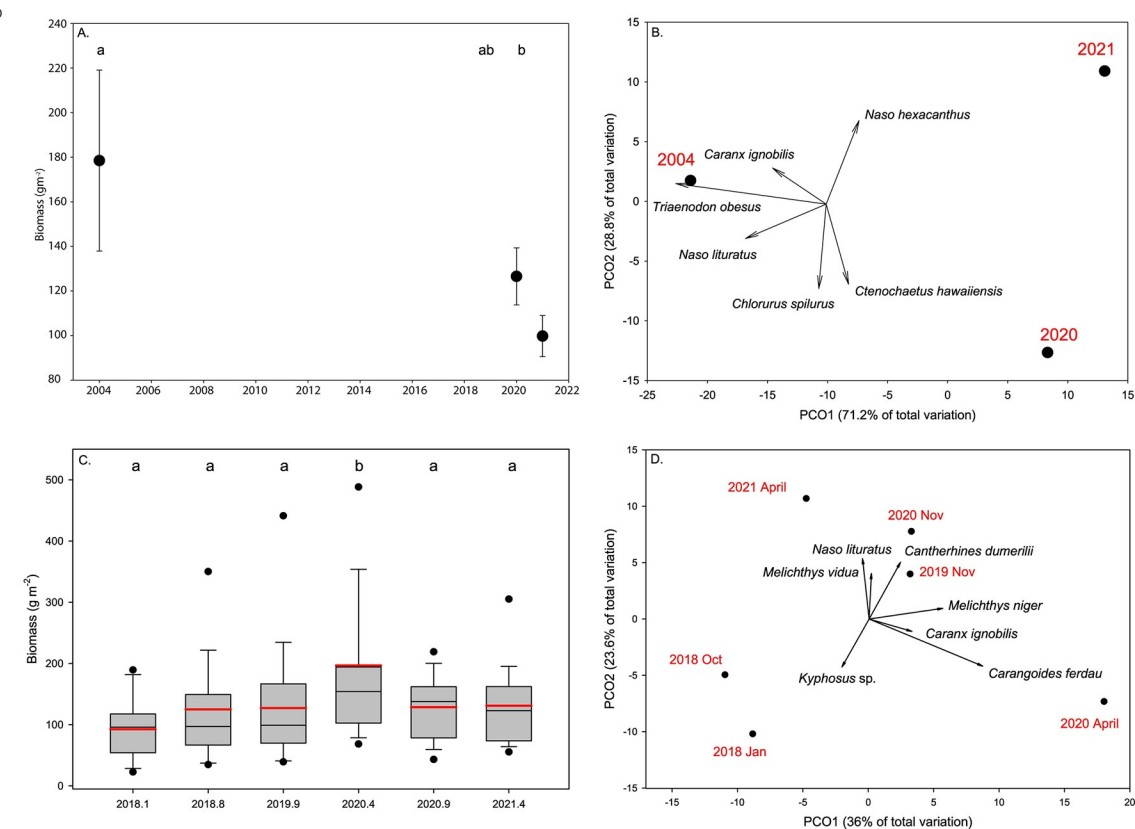

**Fig 3. Changes in the fish community at Molokini.** Comparisons of fish biomass at all trophic levels among sampling periods from (A) long-term surveys and (C) DAR surveys. Box plots showing median (black line), mean (red line), upper and lower quartiles, and 5th and 95th percentiles. Principal coordinates analysis of fish assemblage composition based on biomass (g m⁻²) by sampling period for (B) long-term surveys and (D) DAR surveys. Data were ln(x+1)-transformed prior to analyses. Vectors are the relative contribution and direction of influence of fish taxa to the observed variation among sampling periods (Pearson Product movement > 0.5).

p < 0.001) except for January and November 2018 (R = 0.03, p = 0.06), November 2019 and November 2020 (R = 0.023, p = 0.077), and November 2019 and April 2020 (R = 0.017, p = 0.124). However, these R values are all low and show limited differences in overall assemblage structure among time periods.

The April 2020 sampling period was well-separated in ordination space from the other sampling periods, with jacks (*Uraspis helvola* and *Caranx ignobilis*) most highly correlated with this sampling period (Fig 3D).

## Fish habitat use and occupancy

The daily sum of detections for tagged fishes was negatively correlated with human abundance at Molokini, using data for all species from the whole MPA (both the inside and outside zones) and all hours of the diel cycle (AIC = 27667; Chi-square comparison with null model, DF = 8, LR = 1589, p < 0.001). The pattern was more pronounced when considering only *C. melampygus* from the inside zone during morning hours (Chi-square comparison with null model, DF = 6, LR = 1098, p = < 0.001 Fig 4A). *C. melampygus* showed high utilization of the sheltered interior of the MPA (yellow zone in Fig 1) during the lockdown, but the species was displaced from this habitat during times of high human abundance (morning) starting in December 2020 when tourism resumed (Fig 4B).

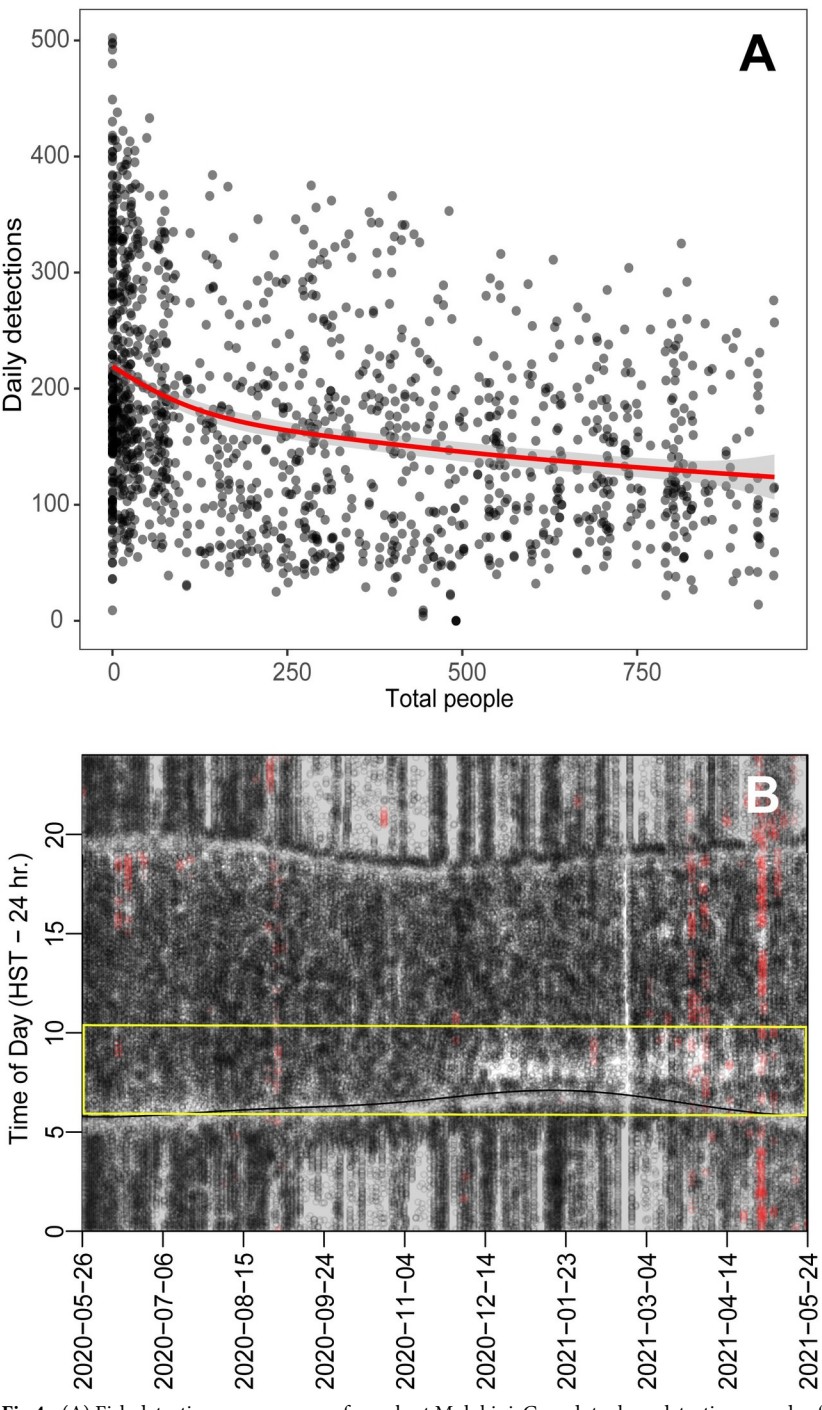

**Fig 4.** (A) Fish detections vs. presence of people at Molokini. Grey dots show detections per day for all *Caranx melampygus* vs. the daily count of people visiting the crater, with darker shade showing higher density. Red line shows GLM fit, grey envelope shows 95% confidence interval. (B) Diel behavior one *C. melampygus* (44 cm fork length) during 2020 and 2021. Y-axis shows diel cycle with time 00:00 (start of diel cycle) at bottom, noon at center and time 23:59 (end of diel cycle). Horizontal curves show dawn and dusk respectively. Dots show detections of *C. melampygus* with black representing inside the crater and red outside the crater. Yellow box indicates morning period of high human abundance. An interpretation guide for this figure is provided in the S1 Fig in S1 File.

## Discussion

Our results demonstrate that there has been a decadal-scale background decline in fish biomass at Molokini, but that some of these losses are driven by the displacement of fishes due to human presence (Fig 4). This suggests that these losses could be recovered with more effective management of tourism, which has been shown to ameliorate tourism impacts in other systems [48]. Multidecadal decreases in fish biomass likely reflect the long-term decrease in fish populations in the region due to fishing, benthic habitat loss, pollution, and other disturbance effects [22]. Since 2004 the fish biomass at Molokini has decreased, but the COVID lockdown and resulting cessation of tourism resulted in a significant rebound of biomass during that time (Fig 4). Given that the species driving this rebound have lifespans and ages at first reproduction of at least several years, the increase in observed biomass did not result from population growth.

Our acoustic tracking data reveal that the biomass recovery during the pandemic resulted from changes in movement patterns as predatory fishes returned to the much quieter and less disturbed MPA (Fig 4). The absence of human disturbance on a timescale of months resulted in fishes moving back into the sheltered interior of Molokini Shoal. The most important species driving these higher biomasses were jacks in the family Carangidae. These are edible and culturally important species that are targeted by fishers, and they are particularly sensitive to human presence. Thus, it is likely that the noise and physical disturbance of large groups of people and boats in shallow water has negative effects on this group of fishes [49].

Upper trophic level species are frequently the first to be removed from ecosystems as a result of human activity [50]. The giant trevally, *C. ignobilis*, is heavily targeted by spearfishers, shorecasters, and often taken in bottomfish and nearshore troll fisheries. *C. ignobilis* drove the changes in biomass during the lockdown (Fig 4). The closely related species *C. melampygus* had higher habitat use of the inside of the crater during the lockdown, and showed displacement behavior once tourism resumed (Fig 4). This displacement from shallow habitat is consistent with a prior study in Molokini Shoal [40], and suggests that high tourism levels may cause some fishes to leave shallow sheltering habitats, potentially exposing them to higher predation risk. Such displacement may also interfere with critical biological events such as spawning, foraging, and resting. *C. melampygus* and *C. ignobilis* are summer spawners [51], so the summertime peak in tourism visitation to Molokini could further reduce population viability for these species.

The displacement of predators in response to human presence altered fish community structure at Molokini in a manner akin to low intensity fishing that affects only the most aggressive and predatory species [52], in contrast to fishing that also degrades herbivory [53, 54]. Predators have diverse ecosystem roles including both consumptive and non-consumptive effects, and the loss of predators can reduce the resistance and resilience of ecosystems to perturbation [55]. In a terrestrial system, the presence of humans caused foraging activity to decrease for predators and increase for prey species [56]. Ongoing human presence can therefore result in long-term displacement of predators from a system, which could then lack the functions provided by this functional group, including mediation of landscape patchiness [57], shoaling of prey species [58] and herbivory [53, 54].

In addition to the impacts of human disturbance on fishes, the level of tourism at Molokini also appears to negatively affect the visitors themselves. A 2011 study [59] found that 67% of visitors felt crowded during their trip and wished to see no more than ~16 boats at one time at Molokini, but this number was exceeded on over 20% of trips to the MPA. Two-thirds of visitors surveyed supported actions that would reduce visitor numbers at Molokini (e.g., limit number of boats and/or people). Visitors expectations for interpretation (e.g., about reefs, history, culture) were not always met [60]. Overall, these findings suggest that Molokini is being used over its capacity, and management is needed to improve visitor experiences.

The impacts of tourism at Molokini are likely to apply to other locations, as MPAs often attract high levels of visitation. No-take MPAs make up <0.5% of nearshore waters in Hawaiʻi [22] but host a disproportionate tourist volume (e.g., Hanauma Bay hosts ~1,000,000/year, Molokini >300,000/year, Kealakekua ~ 36,000/year). As a result, these MPAs receive higher human presence impacts than other less-visited areas, despite the absence of fishing. Management of tourism should include relevant biological research, clear and well-enforced rules, adaptive management, and broad stakeholder involvement [2]. Successful management of tourism requires specific objectives with measurable indicators and outcomes, and identification of threats coupled with strategies to keep impacts below target levels. Strategies to manage the impacts of tourism include legal designation of sites as protected areas, prohibition or limitation of harmful activities, zoning for different levels of protection, permits for tour operators, and limits on visitation [35]. Hanauma Bay on Oʻahu is one of the most-visited MPAs in the world, and has undergone a series of management changes to address excessive visitor volume, including weekly rest days when the park is closed, a ban on fish feeding, a reservation system, mandatory visitor education, and an increase in fees. The series of management actions resulted in a reduction from >8,000 visitors per day in the 1980s to 1,000 per day in 2022 (https://www.honolulu.gov/parks-hbay/2016-09-01-18-10-39/history/timeline.html). Holistic tourism management is necessary throughout Hawaiʻi's waters, particularly in MPAs.

Achieving the goals of Hawaiʻi's Holomua Initiative will require further controls on human impacts, including controls on tourism impacts. Understanding the impacts of non-consumptive uses such as tourism is critical to statewide marine spatial planning, improving effectiveness of existing MPAs, and implementing a statewide network of MPAs. As part of this process, it is necessary to think strategically about the scale and configuration of tourism in Hawaiʻi to optimize earnings and employment without damaging the ecosystem. While staying below a target level of impact, economic benefit can be maximized when focusing on higher margins and lower volumes of tourism [34]. The number of people and the capacity of vessels were both significant predictors of fish displacement from the inside of Molokini Crater. Therefore, managers should consider the total number of visitors as well as the size and capacity of tour boats.

The COVID pandemic caused a strong negative perturbation in human presence at one of the most densely visited tourist sites in Hawaiʻi. We were able to use this natural experiment to demonstrate significant increases in fish biomass and habitat use during the period of human absence, indicating that the business-as-usual conditions of high tourism alter community structure by displacing predatory fishes to deeper environments. As Hawaiʻi formulates marine management plans and undertakes the Sustainable Hawaiʻi Initiative, lessons from Molokini can inform managers and facilitate effective plans.

## Supporting information

**S1 File.**
(DOCX)

**S2 File.**
(JPG)

## Acknowledgments

Field work was conducted by Kristy Stone, Adam Wong, Arthur Wong, Cole Peralto, Tatiana Martinez, Hanalei Silva, Gagan Lally, Matt Chauvin, Donna Brown, Corbin Haliʻimaile Iaea, Dean Tokishi and Kris Billeter. Vessel support was provided by captain Bryce Rohrer and

ProDiver Maui (with support from captains Keone Laepaʻa and Jennifer Meyer). We thank Grace Chiu for suggestions on statistical analyses and Stephen Scherrer for help with coding. In kind support to the project was provided by Rob and Helena Weltman, Bob and Debbie King, Nathan and Cindy Kellogg, Laura Stokes, Don Domingo, Donna Brown, Chris Olsten, Bryan Jaynes and Jeff Milisen. Logistical support was provided by the Maui Nui Marine Resources Council, the Kahoolawe Island Reserve Commission, the University of Hawaiʻi Marine Center, and the Nature Conservancy of Hawaiʻi. Administrative support was provided by the School of Ocean and Earth Science and Technology, the University of Hawaiʻi Office of Research Services, and the VIMS Office of Sponsored Programs, as well as Chris Sabine, Elton Hasegawa, Cindy Forrester, Grace Tisdale, and Elizabeth MacAleese.

## Author Contributions

**Conceptualization:** Kevin C. Weng, Alan M. Friedlander, Russell T. Sparks.

**Data curation:** Kevin C. Weng, Alan M. Friedlander, Laura Gajdzik, Whitney Goodell, Russell T. Sparks.

**Formal analysis:** Kevin C. Weng, Alan M. Friedlander, Laura Gajdzik, Whitney Goodell, Russell T. Sparks.

**Funding acquisition:** Kevin C. Weng, Russell T. Sparks.

**Investigation:** Kevin C. Weng, Alan M. Friedlander, Laura Gajdzik, Whitney Goodell, Russell T. Sparks.

**Methodology:** Kevin C. Weng, Alan M. Friedlander, Laura Gajdzik, Whitney Goodell, Russell T. Sparks.

**Project administration:** Kevin C. Weng, Russell T. Sparks.

**Resources:** Kevin C. Weng, Alan M. Friedlander, Russell T. Sparks.

**Supervision:** Kevin C. Weng, Alan M. Friedlander, Russell T. Sparks.

**Validation:** Kevin C. Weng, Alan M. Friedlander, Laura Gajdzik, Whitney Goodell, Russell T. Sparks.

**Visualization:** Kevin C. Weng, Alan M. Friedlander, Laura Gajdzik, Whitney Goodell, Russell T. Sparks.

**Writing – original draft:** Kevin C. Weng, Alan M. Friedlander, Laura Gajdzik, Whitney Goodell, Russell T. Sparks.

**Writing – review & editing:** Kevin C. Weng, Alan M. Friedlander, Laura Gajdzik, Whitney Goodell, Russell T. Sparks.

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
