## [Decision Letter · Decision Letter 0]

25 Aug 2022

PONE-D-22-19263Decreased tourism during the COVID-19 pandemic positively affects reef fish in a high use marine protected areaPLOS ONE

Dear Dr. Weng,

Thank you for submitting your manuscript to PLOS ONE. After careful consideration, we feel that it has merit but does not fully meet PLOS ONE’s publication criteria as it currently stands. Therefore, we invite you to submit a revised version of the manuscript that addresses the points raised during the review process.

Dear Dr. Weng,

Thank you for submitting your manuscript PONE-D-22-19263 "Decreased tourism during the COVID-19 pandemic positively affects reef fish in a high use marine protected area" to Plos One.

I have now received the report from three reviewers. As you will see they all enjoyed your work and found it interesting and well conducted. However, before to be ready for publication they all recommended some minor revisions. They provided constructive comments and I will be happy to consider a revised version of your work for publication if you can address all these minor comments from reviewers.

Kind regards

Johann

We look forward to receiving your revised manuscript.

Kind regards,

Johann Mourier, Ph.D.

Academic Editor

PLOS ONE

Journal Requirements:

Additional Editor Comments:

Dear Dr. Weng,

Thank you for submitting your manuscript PONE-D-22-19263 "Decreased tourism during the COVID-19 pandemic positively affects reef fish in a high use marine protected area" to Plos One.

I have now received the report from three reviewers. As you will see they all enjoyed your work and found it interesting and well conducted. However, before to be ready for publication they all recommended some minor revisions. They provided constructive comments and I will be happy to consider a revised version of your work for publication if you can address all these minor comments from reviewers.

Kind regards

Johann

Reviewers' comments:

Reviewer's Responses to Questions

**Comments to the Author**

1. Is the manuscript technically sound, and do the data support the conclusions?

Reviewer #1: Yes

Reviewer #2: Yes

Reviewer #3: Yes

2. Has the statistical analysis been performed appropriately and rigorously? 

Reviewer #1: No

Reviewer #2: No

Reviewer #3: Yes

3. Have the authors made all data underlying the findings in their manuscript fully available?

Reviewer #1: Yes

Reviewer #2: Yes

Reviewer #3: Yes

4. Is the manuscript presented in an intelligible fashion and written in standard English?

Reviewer #1: Yes

Reviewer #2: Yes

Reviewer #3: Yes

5. Review Comments to the Author

Reviewer #1: Manuscript: Decreased tourism during the COVID-19 pandemic positively affects reef fish in a high use marine protected area

This paper examined if restrictions due to the COVID-19 pandemic (area closure and low/no tourist activities) influenced fish behavior in a high use no-take marine protected area (MPA) in Hawai`i. The authors found an increase in fish biomass, particularly from predatory species that use shallow habitats during the COVID restrictions in 2020. When tourism resumed, both biomass and habitat use returned to pre-pandemic levels. The authors concluded that managing non-consumptive uses, especially in heavily visited MPAs, should be considered for sustainability of these ecosystems. Overall, this paper was well written, and it is easy to read. The COVID-19 pandemic provided an ideal natural experiment in which scientists could answer a wide range of behavioral and ecological questions. I believe the authors presented a very important and interesting story, and their overall message is critical. We also need to pay attention to the effect of non-consumptive human activities to maintain and restore healthy ecosystems. I don’t have any major comments, but there are several minor issues that they need to be addressed first. Below are some comments that I hope are helpful in strengthening the paper.

Page 3, lines 40-45: Non-consumptive effects are defined in the abstract, but the definition is not clear in the intro. Also it is important to explain how these non-consumptive effects alter species composition and abundance.

Page 3, line 58: Not sure what you mean by “more pristine biomass”

Page 3, lines 60-61: Are these numbers referring only to tourist related activities?

Page 4, lines 74-75: This sentence is not clear. Maybe you can talk about number of visitors instead of margin styles…

Page 5, line 94: “allowed trolling only on the outside.”

Page 6, lines 115-116: There is very little mention of habitat use in the intro. What is the relation between habitat and fish biomass? How do fish use certain habitats? Are there studies contrasting habitat use patterns of fish across gradients of human pressure? Do we have information of fish habitat use in pristine (semi-pristine) environments? We (the reader) need more information and background to justify the hypothesis and perhaps some predictions on what you would expect to find.

Page 5, line 119: “Insights gained from this study may inform future management strategies”

Page 7, lines 144-145: Not sure that I understand the difference

Page 8, lines 155-156: “identified all fishes visible within 2.5 m to either side of the centerline (125 m2 transect area)”

Page 8, line 158: Remove “Yet”

Page 8: lines 154-162: It would be interesting to have some knowledge of where these transects were conducted so

that we have some sense of what was the total area that was covered.

Page 9, line 174: “if a fish was inside or outside of the crater”

Page 9, line 186: what species and how many individuals from each species were tagged? Why these species were important to answer your question/hypothesis? I think this is something that the intro/methods should explore more. Are you tagging only predatory fishes? Why? In the results there is only information of C. melampygus.

Page 10, lines 196-197: The model needs more explanation. What is Y in the equation (I am assuming is human abundance, but it can be clearer in the text)? Why using cos? how did you compare and selected models? How you assess the fit and model performance? What software and/or R functions did you use for this procedure?

Page 10, line 204: You only included year as a factor in the model or you also account for other factors (depth, temperature, season, etc)? It would be important to include other factors available in the model that could influence fish distribution and abundance. Also, did you use data from both fish survey protocols for this? Yes/No, why?

Page 10, line 205: why Poisson distribution? Can you expand on this?

Page 11, line 214: What is a sampling period? Also, it would be good to have an idea of what a full/complete model looks like and to all the candidate models that you consider, as well as the model that was selected and the evidence for that specific model.

Page 11, line 216: Why? It seems to me that transect is the sampling unit that you are using so not sure why it is being considered a random effect in the models?

Page 11, lines 226-228: This needs to be further explained. How exactly did you combine these data?

Page 11, lines 233-234: What not using a predictor of human activity that could combine all these 3 variables instead?

Page 13, lines 266-267: giant travally and whitetip reef shark are not top predators, they are mesopredators. Giant travally is a large mesopredator and whitetip reef shark is a small mesopredator. It would be important to adecuately define what a top predator is as there is a lot of evidence that suggest that both species are actually mesopredators rather than top predators.

Page 14, line 278: Is this refers to the biomass of all species or predator species or top predators (based on your text)? This is not clear.

Page 14, line 290: More information about the models is needed. i suggest having a table with the full model and candidate models that were evaluated as well as their support

Page 15, line 299: This panel is confusing. There is a lot going on and the patterns are not clear.

Page 16, line 322: “resulting cessation of tourism resulted in a significant rebound of biomass during April 2020”

Page 18, line 370: Maybe you can move this text at the beginning: “Understanding the impacts of non-consumptive uses such as tourism is critical to statewide marine spatial planning, improving effectiveness of existing MPAs, and implementing a statewide network of MPAs.”

Reviewer #2: Review of manuscript titled “Decreased tourism during the COVID-19 pandemic positively affects reef fish in a high use marine protected area”, submitted for publication in the journal PlosOne. This study examines the effects of the high intensity non-extractive tourism on fish community composition in the shallow waters of an MPA in Hawaii. The authors utilise the changes in tourism numbers associated with the COVID-19 pandemic as a factor in the experiment, finding that a decrease in tourist numbers was associated with a short term increase in mobile higher order predators (jacks) in shallow waters. The authors link this with behavioural displacement, associated with avoidance behaviours at times and locations when visitor use is high. This is a well thought out and executed study with some very interesting results that have repercussions for conservation management. This joins a growing body of literature that use utilised the COVID pandemic as a case study to examine the effects of sudden changes in human usage patterns on marine ecosystems. I feel that the study contributes to the wider literature and is suited to the scope and audience of PlosOne. I would recommend it for publication with some minor corrections. I would be happy for the editor to asses whether my comments have been addressed upon resubmission.

Specific comments are as follows:

Abstract

- Recommend that the authors substitute the word ‘consumptive’ with ‘ extractive’ throughout the manuscript. Consumptive implies that a resource is eaten or consumed. Fishing activity isn’t always consumptive, but does result in extraction from a community. My experience is that extractive is a more commonly used term in the literature.

- Ln 30: reword sentence to read ‘change community composition and habitat usage at local scales…’

- Ln 31: I don’t think your findings are strong enough to say that you have evidence that tourism ‘can compromise ‘ ecosystem function. Please reword to ‘could impact’ ecosystem function.

Introduction

- As above, recommend changing ‘consumptive’ to ‘extractive’ throughout.

- Ln 42: Recommend changing ‘humans’ to ‘human activity.’ Fish won’t necessarily modify behaviour to humans themselves, but rather the activities that they conduct. As an example, vessel movement may alter behaviour.

- Ln 57: Change to ‘negative impacts that can occur as a result of tourism.’ Not all tourism is detrimental or modifies behaviour in a significant way.

- Ln 57: change ‘abundance’ to ‘activity.’

- Ln 58: change to ‘re-establish optimal habitat use.’ The fact that the fish are still around indicates that they are still utilising the habitat, just not in their preferred way.

- Ln 67-84: Some of the content of these two paragraphs is likely better suited to your discussion. At the moment your intro is quite lengthy.

- Ln 111: Change to ‘displaced from inside the’

Methods

- Ln 154: What is the experience level of divers/observers conducting surveys and how is quality control ensured? Were divers/observers consistent between temporal surveys? Is there the potential that there was any observer bias associated with the 2004-2020/21 survey differences?

- Ln 155: Does this include cryptic species?

- Ln 169: How long was the telemetry data recorded for (i.e. start and finish dates)?

- Ln 186: Please provide more detail on species tagged in this experiment. Its as much of an interest to know which species didn’t modify behaviour, as those that did.

- Ln 204: Is there supposed to be a sub heading for this paragraph, as it doesn't appear to fit under the previous heading

- Ln 207: Did you consider running a PERMANOVA to statistically examine shifts in community structure? Given that you state that there was a shift in community composition as one of your conclusions, I feel a statistical test is warranted. When combined with a CAP analysis, would enable you to identify what species are statistically responsible for the shifts.

- Ln 213: As above with regards to the PERMANOVA and CAP analysis.

Discussion

- While interesting, the discussion is highly focused on management recommendations. While this should definitely be included in your discussion, I feel that it could be refocused a little to also include discussion of the potential ecosystem effects of your findings. In particular, I would like to see some discussion of the potential trophic and ecosystem function effects of your findings. Your primary effects have been on higher order mobile predator species, ironically the same groups that tend to be highly targeted by fishing activity. As such, high intensity tourism may well be having similar effects at local scales as low intensity fishing (i.e. fishing effort at a level that primarily only effects higher order groups). This is actually a really interesting finding and worth discussing in more detail.

- Ln 332: Change to ‘on this group of fishes.’

- Ln 387: I disagree with this statement. You haven't examined habitat at all. What you have shown is that high tourism can affect the space usage of some species, particularly higher order predators. This could have repercussions for trophic links and ecosystem function akin to the effects of fishing (which also tends to target higher order group).

Reviewer #3: This paper used the reduction in tourism caused by the COVID-19 pandemic as a natural experiment to assess the impact of tourism on fish communities inside the Molokini no-take marine protected area (MPA) in Hawai’i. They found that during the tourism shut down, fish biomass increased and predatory fishes increased the use of shallow habitats. However, these changes were temporary as species were displaced from the MPA when tourism resumed. Generally, I think this paper is very interesting and takes advantage of a unique set of circumstances to provide an insight into the impacts of human activities that are difficult to quantify. The implications of this study are important, as they suggest that tourism can displace species that are targeted by fishes from areas that are protected from fishing.

I have a few minor comments for the authors to consider. Mainly I was unable to review the Results in detail as the figures are in a very low resolution and very hard to read.

Introduction:

Line 62: This is an interesting example, but perhaps the authors could be more specific about the drivers of this increase in fish abundance (i.e. Lecchini et al. (2021) found beavhioural changes that lead to increased measures of fish biomass instead of actual changes in population density). A few linking words could also help the flow of this paragraph. Maybe something like “In French Polynesia, this decrease in tourism resulted in an associated change in fish community structure, where surveys revealed a short term increase in fish biomass in the absence of tourism”.

Line 65: I would leave this last line a bit more open to highlight a knowledge gap rather than make predictions/hypothesis in the middle of the introduction.

Methods: generally these seem appropriate for the questions being asked. The sampling gap between 2004 - 2021 is not ideal, but I understand that sampling is rarely perfect when using historical data sets. A table of the tagged species would be useful. This would help the reader understand sample size, what species have been tagged, and is meant exactly by the term “predatory fishes”. Depending on how many different species were tagged, the differences in their ecology (home ranges etc) and the degree to which they are targeted by fishers, it could be interesting to separate these species in the analysis.

Results: It is honesetly very hard to comment on the results because I cannot read the graphs.

Discussion:

I think the Discussion makes som excellent and interesting points. I do wonder whether the impact of displacing predators from shallow reefs cascades down to lower trophic levels. There are a few examples from the terrestrial literature of humans changing predator behaviour, which subsequently benefits lower trophic levels (e.g. Suraci, Justin P., et al. "Fear of humans as apex predators has landscape‐scale impacts from mountain lions to mice." Ecology Letters 22.10 (2019): 1578-1586.). Could be something to consider adding into the discussion.

Line 305-313: the first paragraph of the Discussion seems more suited to the Introduction as there are no results or discussion of these findings. I would suggest starting with paragraph 2 (which is a really interesting).

Line 332: Line 332- The authors make a good point about noise disturbing the fish (although some references would be good here). In addition to noise, the presence of humans could also displace fish. If fish encounter spearfishers during their movements, they can learn that humans represent a potentially lethal threat, especially species that are targeted by these fishes, and flee from these fishers

These reference could be useful:

Goetze, Jordan S., et al. "Fish wariness is a more sensitive indicator to changes in fishing pressure than abundance, length or biomass." Ecological Applications 27.4 (2017): 1178-1189.).

Januchowski-Hartley, Fraser A., et al. "Fear of fishers: human predation explains behavioral changes in coral reef fishes." PLoS One 6.8 (2011): e22761.

Januchowski-Hartley, Fraser A., Kirsty L. Nash, and Rebecca J. Lawton. "Influence of spear guns, dive gear and observers on estimating fish flight initiation distance on coral reefs." Marine Ecology Progress Series 469 (2012): 113-119.

Line 352: should this be “our findings suggest” ?

Line 354: what aspect of visitor education could be enhanced?

Line 368: be specific about what is meant by the “health” of the MPA

6. PLOS authors have the option to publish the peer review history of their article (what does this mean?). If published, this will include your full peer review and any attached files.

Reviewer #1: **Yes: **Mario Espinoza

Reviewer #2: No

Reviewer #3: No

---

## [Author Response · Author response to Decision Letter 0]

3 Feb 2023

Please see "Response to Reviewers"

---

## [Decision Letter · Decision Letter 1]

14 Mar 2023

Decreased tourism during the COVID-19 pandemic positively affects reef fish in a high use marine protected area

PONE-D-22-19263R1

Dear Dr. Weng,

We’re pleased to inform you that your manuscript has been judged scientifically suitable for publication and will be formally accepted for publication once it meets all outstanding technical requirements.

Kind regards,

Andrea Belgrano, Ph.D.

Academic Editor

PLOS ONE

Additional Editor Comments (optional):

Based on my evaluation of the revised manuscript and the positive comments on the revised manuscript by Reviewer #2, my recommendation is to accept the manuscript in its present form.

Reviewers' comments:

Reviewer's Responses to Questions

**Comments to the Author**

1. If the authors have adequately addressed your comments raised in a previous round of review and you feel that this manuscript is now acceptable for publication, you may indicate that here to bypass the “Comments to the Author” section, enter your conflict of interest statement in the “Confidential to Editor” section, and submit your "Accept" recommendation.

Reviewer #2: All comments have been addressed

2. Is the manuscript technically sound, and do the data support the conclusions?

Reviewer #2: Yes

3. Has the statistical analysis been performed appropriately and rigorously? 

Reviewer #2: Yes

4. Have the authors made all data underlying the findings in their manuscript fully available?

Reviewer #2: Yes

5. Is the manuscript presented in an intelligible fashion and written in standard English?

Reviewer #2: Yes

6. Review Comments to the Author

Reviewer #2: I am satisfied that the authors have addressed all of my comments. The authors have done an excellent job putting this together and I wish them all the best.

7. PLOS authors have the option to publish the peer review history of their article (what does this mean?). If published, this will include your full peer review and any attached files.

Reviewer #2: No

---

## [Editor Report · Acceptance letter]

21 Mar 2023

PONE-D-22-19263R1 

Decreased tourism during the COVID-19 pandemic positively affects reef fish in a high use marine protected area 

Dear Dr. Weng:

I'm pleased to inform you that your manuscript has been deemed suitable for publication in PLOS ONE. Congratulations! Your manuscript is now with our production department. 

Kind regards, 

on behalf of

Dr. Andrea Belgrano 

Academic Editor

PLOS ONE